# Inferring structure and parameters of stochastic reaction networks with logistic regression

**Boseung Choi**[1,2,3⊘], **Hye-Won Kang**[4⊘], **Grzegorz A. Rempala**[3⊘]*

**1** Korea University Sejong Campus, Sejong, South Korea, **2** Institute for Basic Science, Daejeon, South Korea, **3** The Ohio State University, Columbus, Ohio, United States of America, **4** University of Maryland, Baltimore County, Baltimore, Maryland, United States of America

⊘ These authors contributed equally to this work.
* rempala.3@osu.edu

## Abstract

Identifying network structure and estimating reaction parameters remain central challenges in modeling chemical reaction networks. In this work, we develop likelihood-based methods that use multinomial logistic regression to infer both stoichiometries and network connectivity from full time-series trajectories of stochastic reaction systems. When molecular counts for all species are observed, stoichiometric coefficients can be recovered provided that each reaction occurs at least once during the sampling window and has a unique stoichiometric vector. We illustrate the proposed regression approach by recovering the network structure in three stochastic models involving catalytic interactions in open networks—namely, the Togashi–Kaneko model, a heat-shock protein network model, and a Susceptible–Infected–Recovered (SIR) epidemic model. We then demonstrate the practical value of the method using synthetic epidemic data designed to mirror key features of the COVID-19 outbreak in the Greater Seoul area of South Korea. In this example, we analyze an SIR network model with demographic effects and address partial observability—specifically, the fact that only infection counts are observed—by combining Bayesian logistic regression with differential-equation modeling. This integrated framework enables reliable recovery of core SIR parameters from a realistic, COVID-like synthetic trajectory of disease prevalence. Overall, our results show that relatively simple likelihood-based tools, such as logistic regression, can yield meaningful mechanistic insight from both synthetic systems and data that reflect real-world epidemic dynamics.

## Introduction

Identifying network structures from time-series trajectories is a fundamental challenge across scientific disciplines, including systems biology, epidemiology, and chemical kinetics. A variety of methods have been proposed to address this problem, depending on factors such as the completeness of the observations, the temporal resolution of the data (continuous vs. discrete), and the presence of noise.

**Data availability statement:** The simulated time-series datasets generated for the models considered in this study are available on Zenodo at [https://doi.org/10.5281/zenodo.17658259]. All analyses were conducted in R (version 4.4.2). The complete set of custom code developed by the authors is also available in the same repository/DOI. The datasets used in the project are available as follows. • Data for Fig 1. Times series data of the TK model. https://github.com/cbskust/Reaction.Identification/blob/v1.0.0/TK_model_Multinomial_Case1.csv • Data for Fig 2. Time series data for HSR model. https://github.com/cbskust/Reaction.Identification/blob/v1.0.0/HeatShock_model_Multinomial_Case1.csv • Data for Fig 3. Time series data from the SIR model with demography. https://github.com/cbskust/Reaction.Identification/blob/v1.0.0/SIR_model_Multinomial.csv • Data for Fig 4. Synthetic prevalence of COVID-19 in Seoul, South Korea, from Oct. 17, 2020, to Jan. 24, 2021. https://github.com/cbskust/Reaction.Identification/blob/v1.0.0/COVID-19.csv.

**Funding:** This research was partially supported by The Ohio State University HEALMOD Initiative. In addition, Bosung Choi was supported by the Basic Science Research Program through the National Research Foundation of Korea (NRF), funded by the Ministry of Education (grant no. RS-2023-00245056), and by the Government-wide R&D to Advance Infectious Disease Prevention and Control program (grant no. HG23C1629). No additional external funding was received for this study.

**Competing interests:** The authors have declared that no competing interests exist.

In deterministic mass action kinetics, different reaction networks may produce identical dynamics, and model parameters may not be uniquely identifiable [1]. *Ordinary differential equations* (ODEs) with polynomial right-hand sides are commonly used to select relevant terms and estimate parameters from time-series data, employing regression [2–4] and machine learning techniques [4]. These methods have been extended to infer reaction networks under mass action kinetics [5], and further adopted for fully or partially observable species using Bayesian approaches [6].

A similar approach has been developed in the stochastic setting under the assumption that all species and all reaction events are fully observable. In that framework, polynomial propensity functions up to a specified degree are taken as basis functions, and the parameters within these functions are inferred by maximizing the likelihood [7]. It has been shown that both network structure and reaction parameters can be uniquely identified when transition rates over a sufficiently large state space, or a sufficient number of full trajectories, are available [8].

Although full observability enables network identifiability results, the task of parameter estimation poses further challenges. When the network structure is known, parameter estimation still requires time-series data. Under full observability, Bayesian inference via *Markov Chain Monte Carlo* (MCMC) methods—using reversible-jump algorithms or block updating—is effective [9]. For partially observed systems, these methods can be adapted [9], and uniformization techniques can be used to approximate conditional distributions [10]. MCMC methods incorporating distributed time delays can further accommodate partial observations by simultaneously inferring both system parameters and the delay distribution [11].

In this paper, we study stochastic reaction networks in a well-mixed environment, where species undergo creation, decay, and interactions over time. In contrast with [8] and [7], our aim is to develop simple, discrete, and scalable likelihood-based methods for network inference and parameter estimation in settings where full trajectory data may not be available. Instead, we assume that the selected jump types are observed for at least some species of interest, with only limited structural information available for any unobserved species. We also assume that the network contains at least one pure-birth reaction. This latter assumption substantially simplifies the structural identification problem studied in [8], ultimately leading to a convenient logistic-regression formulation.

As in [8], our approach is designed for reaction networks in which each reaction type has a unique stoichiometric signature. A formal statement of this assumption is provided in the theorem included in the Supplement S5 file in supporting information. While extensions to more general cases are possible, we focus here on the simplest setting. We also offer practical guidance on how to proceed when this condition is only mildly violated, as illustrated in one of our examples involving a heat-shock model.

In what follows, we organize the network inference task into two sequential stages:

1. **Network structure identification**: In the first stage, we aim to recover the underlying reaction network by identifying which species influence the occurrence of particular reactions. We show that logistic regression can be applied directly to fully observed trajectory data (and, in fact, the embedded discrete chain alone suffices), allowing us to determine which species' abundances are associated with specific reaction events.
2. **Parameter estimation**: After establishing the network structure, the second stage aims to estimate the *kinetic reaction rate constants*, which characterize the magnitude and frequency of interactions within the system. For this purpose, we again employ a logistic regression framework, enhanced with an offset correction, to infer the reaction rate parameters from the same trajectory data.

Building on this two-step approach, we structure the paper as follows. In the Methods section, we introduce the formalism of chemical reaction networks and present a general framework that uses logistic regression for network inference and Bayesian inference for the logistic regression model.

In the Results section, we first demonstrate the structure-inference step using three numerical examples: the *Togashi–Kaneko* (TK) model—an autocatalytic chemical reaction network [12]; a heat-shock response model [13]; and a stochastic *Susceptible–Infected–Recovered* (SIR) model with birth and death processes, widely used in epidemiology [14].

We then turn to parameter estimation based on real-world data. Specifically, we apply our methods to daily new infection count data mirroring the early stages of the COVID-19 epidemic in the Greater Seoul area. Using SIR model with demographic dynamics, we demonstrate how this widely used framework offers a practical approach for estimating key epidemiological parameters, including transmission and recovery rates.

In the Discussion section, we summarize our findings, comment on the performance of the proposed method, and highlight several directions for future work.

By decomposing the problem into two distinct steps—network structure inference followed by parameter estimation—we propose a practical and interpretable approach for reconstructing reaction networks from time-series data. Our findings show that even simple likelihood-based techniques, when applied appropriately, can reliably uncover underlying network structures and produce plausible parameter estimates in both synthetic and real-world settings.

## Methods

### Network structure identification

We consider a chemical reaction network (CRN) involving $m$ chemical reactions and $s$ chemical species. Denote the $s$-dimensional vector of species by $A$ with its $i$-th species component denoted $A_i$, for $i = 1, 2, \cdots, s$. The stoichiometric coefficient $\nu_{ik}$ ($\nu'_{ik}$), for $i = 1, 2, \cdots, s$, $k = 1, 2, \cdots, m$, represents the number of molecules of species $A_i$ that is consumed (or produced) in the $k$-th reaction. The stoichiometric vectors $\nu_k$ and $\nu'_k$ are defined as $s$-dimensional vectors whose $i$-th components are $\nu_{ik}$ and $\nu'_{ik}$, respectively. Then, the CRN is given as

$$\sum_{i=1}^{s} \nu_{ik} A_i \longrightarrow \sum_{i=1}^{s} \nu'_{ik} A_i, \quad \text{for } k = 1, 2, \cdots, m. \tag{1}$$

The species on the left-hand side of a reaction are often referred to as *reactants*, while those on the right-hand side are referred to as *products*.

We model the *stochastic chemical reaction network* (CRN) in (1) as a continuous-time Markov jump process. For readers interested in background on this formulation, a concise introduction to Markov chain models for CRNs is provided in [15]. We assume that the rate function of the $k$-th reaction is $a_k(A)$, where $a_k$ is a polynomial in species counts and that for some $r$ ($1 \leq r \leq m$) we have $a_r(A) = const$, that is, we have at least one constant rate function. We further assume that

each stoichiometric vector $v'_k - v_k$ corresponds uniquely to a reaction type, so that the total number of reactions $m$ matches the number of distinct stoichiometric vectors. In practice, however, the method remains reasonably robust even when this last condition is mildly violated, as illustrated in the second example in the Results section.

We generate full trajectory data for the stochastic model using Gillespie's *Stochastic Simulation Algorithm* (SSA) [16,17]. These synthetic trajectories form the basis for network inference, initially under the assumption that all species are observable at every reaction event. Notably, large simulation sets are not required: as shown in the Results section, accurate recovery of the network structure can be often achieved with only a modest number of trajectories (e.g., around ten).

Since reaction events are observed at all time points, the stoichiometry of each reaction can be identified from the dataset, provided that all reactions occur at least once during the observation period. Once the stoichiometry is known, the net changes in molecular counts can be determined; however, the presence of catalysts in the reactions cannot be inferred from this information alone, as catalysts do not alter stoichiometry.

To identify the catalysts, we employ a *multinomial logistic regression approach*. Let $X_i$ denote the molecular count of the $i$-th species, and let $X$ denote the corresponding vector of species counts. We assume that the log-ratio of the probability of occurrence of the $k$-th reaction relative to a chosen reference reaction can be expressed as a linear function of some transformation $z(\cdot)$ of species counts. The choice of transformation $z(X_i)$ may vary, provided it is an increasing function of the molecular counts $X_i$. In this study, we use the logarithmic transformation $z(X_i) = \log X_i$. This transformation is canonical in the sense that under certain conditions it makes the embedded Markov chain likelihood equivalent to the multinomial logistic regression likelihood. The detailed correspondence is presented in the theorem provided in the Supplement S5.

The multinomial logistic regression model [18] is then given by

$$\log \frac{P(Y = k|X)}{P(Y = r|X)} = \alpha_k + \sum_{i=1}^{s} \beta_{ki} z(X_i), \qquad \text{for } k = 1, 2, \cdots, m, \ k \neq r. \tag{2}$$

Here $Y$ denotes a categorical random variable indicating the reaction type, with the $r$-th reaction chosen as the reference category in the multinomial logistic regression model. Each stoichiometric type is treated as a nominal category. For each observed reaction event, we pair the molecular counts of all species immediately preceding the event with the corresponding stoichiometric category. Using this combined dataset, we fit a multinomial logistic regression model, typically selecting a production reaction ($r$) as the reference category.

The estimated regression coefficients $\beta_{ki}$ in (2) capture how species abundances affect the likelihood of each reaction type. Assuming $z(X_i)$ is an increasing function of the molecular counts $X_i$, the *sign* of each coefficient aids interpretation: a positive value indicates that higher species counts increase the reaction likelihood, suggesting a reactant role, whereas a negative value indicates that higher species counts decrease the reaction likelihood and it therefore not of direct interest in our analysis. Species with no net change, such as catalysts, can still be identified through their positive association with reaction propensity despite unchanged molecular counts. When using the logarithmic transformation $z(X_i) = \log X_i$, the estimated coefficients $\beta_{ki}$ recover the power of $X_i$ in the propensity $a_k(X)$, which also corresponds to the consumed molecular count $v_{ik}$ of the $i$-th species. This framework thus enables systematic detection of reactant species, including catalytic participants. For further details, see Supplement S5.

## Parameter estimation

For parameter estimation, we adopt a slightly more general—and thus more flexible—form of the model in (2) and employ a Bayesian framework, wherein the regression coefficients and any additional unknown terms are jointly inferred through their posterior distributions.

We consider a CRN as described in (1), and assume that a subset of the chemical species is observable—that is, full trajectory data are available for these observable species, while the remaining species are unobserved. For the reactions involving unobserved species, we assume that approximate trajectory information may be available from other sources, such as some auxiliary dynamical models.

We further generalize the model in (2) by assuming that the log-ratio of the probability of the $k$-th reaction, relative to a reference reaction, is a linear function of some transformation $z(\cdot)$ of the observed species counts, augmented by an offset term. The resulting multinomial logistic regression model takes the form

$$\log \frac{P(Y_j = k|X_{\mathcal{O},j})}{P(Y_j = r|X_{\mathcal{O},j})} = \alpha_k + \sum_{i \in \mathcal{O}} \beta_{ki}\, z(X_{ij}) + \text{offset}_{kj}, \ \text{ for } k = 1, 2, \cdots, m, \ k \neq r, \tag{3}$$

where the $r$-th reaction is chosen as the reference category. Here, $Y_j$ denotes the categorical random variable indicating which reaction fires at the $j$-th jump of the process (i.e., the $j$-th observation time) for $j = 1, 2, \cdots, \ell$ where $\ell$ denotes the total number of jumps during the observed time. The quantity $X_{ij}$ denotes the molecule count of species $i$ immediately *before* the $j$-th jump, and $X_{\mathcal{O},j}$ denotes the vector of observed species at this same pre-jump state.

The term offset$_{kj}$ accounts for the contribution of unobserved species to the propensity of the $k$-th reaction at the $j$-th jump. In practice, this offset can be estimated or approximated using a reduced dynamical model or filtering procedure for the unobserved subsystem evaluated at the pre-jump state. Thus, the quantity $P(Y_j = k \mid X_{\mathcal{O},j})$ represents the probability that the $k$-th reaction fires at the $j$-th jump time, conditional on the observed component of the state just before the jump.

Let $\theta$ denote the vector of unknown parameters to be estimated. The likelihood function for the observed data $Y$ can be expressed as

$$L(\theta \mid Y, X_{\mathcal{O}}) = \prod_{j=1}^{\ell} \prod_{k=1}^{m} P(Y_j = k|X_{\mathcal{O},j})^{\mathbb{1}_{\{y_j = k\}}}, \tag{4}$$

where $\mathbb{1}_{\{y_j = k\}}$ is the indicator function and $y_j \in \{1, 2, \ldots, m\}$ denotes the observed outcome of the random variable $Y_j$ at time $j$. The posterior distribution of the parameters, $\pi(\theta \mid Y, X_{\mathcal{O}})$, is then given up to a normalizing constant by

$$\pi(\theta \mid Y, X_{\mathcal{O}}) \propto L(\theta \mid Y, X_{\mathcal{O}})\, \pi(\theta), \tag{5}$$

where $\pi(\theta)$ represents the prior distribution encoding information about $\theta$ before observing the data.

To carry out Bayesian estimation, we employ *Markov Chain Monte Carlo* (MCMC) methods to approximate samples from the posterior distribution $\pi(\theta \mid Y, X_{\mathcal{O}})$. Specifically, we use the *robust adaptive Metropolis* (RAM) algorithm [19], which dynamically adjusts the proposal covariance matrix to achieve an optimal acceptance rate, improving sampling efficiency and reducing the need for manual tuning (see also [20] for a broader discussion of adaptive MCMC). The resulting samples enable computation of posterior summaries and facilitate probabilistic inference for the model parameters. For additional discussion and illustrative examples, see [21].

## Results

### Network structure identification

In the following, we illustrate our network identification approach using three synthetic yet biologically motivated examples of reaction networks: the well-studied *Togashi–Kaneko* (TK) model, a *canonical heat shock response model*—both of which have been previously analyzed in the literature (see, e.g., [12] and [13])—and the *SIR model with demography* [14], which is widely known and considered a foundational workhorse in epidemic modeling.

 

**The Togashi-Kaneko (TK) model.** The TK model consists of a cycle of autocatalytic reactions, along with inflow and outflow reactions. It was first introduced by Togashi and Kaneko [12]. More recently, the long time behavior of the TK model, as well as its generalizations, has was studied in [22]. The reaction network for $m$ species $\{A_i\}_{i=1}^m$ is given by:

$$A_i + A_{i+1} \xrightarrow{\kappa_i} 2A_{i+1}, \quad i = 1, 2, \dots, m, \quad \text{where } A_{m+1} = A_1. \tag{6}$$

$$\varnothing \underset{\delta_i}{\overset{\lambda_i}{\rightleftharpoons}} A_i, \quad i = 1, \dots, m. \tag{7}$$

This model plays a significant role in molecular-level stochastic kinetics, as it exhibits discreteness-induced transitions that are absent in deterministic formulations. Consequently, the TK model may serve as an illustrative test case for network inference using a logistic regression approach.

We specifically consider the case with two species ($m = 2$) to evaluate the method's ability to identify all reacting species (reactants) whose molecular counts influence the reaction rates. Here, the reaction rates are constructed under mass action kinetics. For example when $m = 2$, $a_i(x) = \kappa_i x_1 x_2$, $a_{i+2}(x) = \lambda_i$, $a_{i+4}(x) = \delta_i x_i$ for $i = 1, 2$ where $x = (x_1, x_2)$ and $x_i$ are nonnegative integer values denoting the molecular counts of the $i$-th species.

Our analysis examines two distinct scenarios of fast autocatalytic reactions: one with symmetric reaction rates and another with asymmetric rates between the two species. In the *symmetric* reaction case (Case 1), we assume $\kappa_1 = \kappa_2$, $\lambda_1 = \lambda_2$, and $\delta_1 = \delta_2$. In the *asymmetric* reaction case (Case 2), these parameters are allowed to take distinct values. For each scenario, we generate 10 independent datasets, with representative examples shown in Fig 1. Logistic regression is then applied to each dataset, using the production of species $A_2$ as the reference reaction.

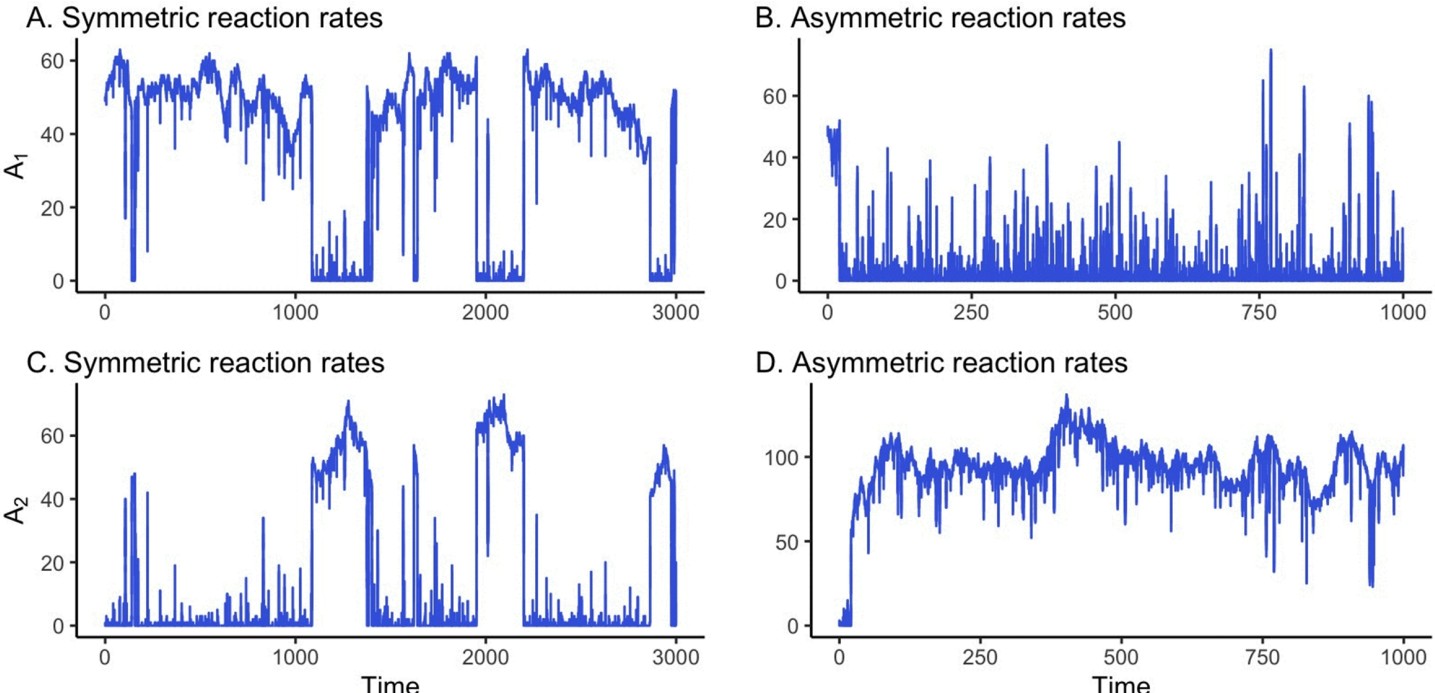

**Fig 1**. **Times series data of the TK model.** A set of representative stochastic trajectories of the TK model with initial conditions $A_1(0) = 49$ and $A_2(0) = 1$. Panels (A,C) illustrate the case of symmetric reaction rates, with $\kappa_i = 200$, $\lambda_i = 0.2$, and $\delta_i = 0.0078$ for $i = 1, 2$. Panels (B,D) depict asymmetric rates, given by $\kappa_1 = 20$, $\kappa_2 = 19$, $\lambda_1 = 2$, $\lambda_2 = 1$, $\delta_1 = 0.078$, and $\delta_2 = 0.03$.

In the symmetric case, stochastic trajectories of the TK model alternate between low- and high-concentration states over time (Fig 1, left). By contrast, with asymmetric reaction rates, species $A_1$ fluctuates around zero while $A_2$ remains at elevated levels (Fig 1, right). Although both species exhibit pronounced fluctuations in molecular counts, they do not exhibit the abrupt transitions between low- and high-concentration states that characterize the symmetric case.

From the regression results—specifically, the estimated coefficients—we can identify when the coefficients for $A_1$ or $A_2$ are statistically significant and positive. Such significance implies that the corresponding reaction is more likely to account for the observed stoichiometric change (i.e., jump type) relative to the reference reaction. For the TK model, the production of $A_2$ serves as the reference. A significantly positive coefficient for a given set of reactants indicates that the associated reaction is more likely to occur when $A_1$, $A_2$, or both are present at elevated levels.

To better control the false positive rate, we apply a stringent criterion for identifying *significant* regression coefficients, requiring one-sided $P$-values below 0.001, which corresponds to $Z$-scores of 3.09 or higher.

For both the symmetric and asymmetric reaction rate cases in the TK model, all reactants involved in first- and second-order reactions are correctly identified, as indicated by their statistically significant positive regression coefficients (Table 1). For the zeroth-order reaction, we confirm that the production of $A_1$ has no reactants in both cases, as the coefficients for $A_1$ and $A_2$ are not significant in the regression (Table 1).

We can also identify statistically significant positive coefficients from the distribution of the $Z$-values. As an example, S3 Fig illustrates the distribution for the symmetric reaction case (Case 1). Notably, the $Z$-values separate clearly into two groups - those exceeding the threshold indicated by the red vertical line and those falling below it.

**The heat shock response (HSR) model.** As our second example of network identification in an open reaction system, we consider the heat shock response (HSR) model, originally developed by Linder and Rempala [13]. This model involves two types of proteins, $P_1$ and $P_2$, as well as gene expression represented by $R_1$.

In the HSR model, the two proteins—heat shock transcription factors—form a positive feedback loop that promotes the production of each other. Additionally, they enhance the expression of heat shock protein genes. The gene product $R_1$ can self-amplify its own expression and regulate one of the transcription factors ($P_2$) through a negative feedback mechanism.

$$\varnothing \xrightarrow{\kappa_1} P_1, \quad \varnothing \xrightarrow{\kappa_2} P_2, \quad \text{(Natural production)}$$

$$P_2 \xrightarrow{\kappa_3} P_1 + P_2, \ P_1 \xrightarrow{\kappa_4} P_1 + P_2, \ R_1 \xrightarrow{\kappa_5} 2R_1, \ \text{(Catalyzed production)}$$

$$P_1 \xrightarrow{\kappa_6} R_1, \quad P_2 \xrightarrow{\kappa_7} R_1, \quad R_1 \xrightarrow{\kappa_8} P_2, \quad \text{(Conversion)}$$

$$P_1 \xrightarrow{\kappa_9} \varnothing, \quad P_2 \xrightarrow{\kappa_{10}} \varnothing, \quad R_1 \xrightarrow{\kappa_{11}} \varnothing, \quad R_1 + P_2 \xrightarrow{\kappa_{12}} \varnothing, \quad \text{(Degradation)}$$

This model features a more complex reaction network than the two-species TK model, comprising three species and twelve reactions. A distinctive feature is the presence of two pairs of reactions with identical stoichiometries: the natural

**Table 1. Species identification in the TK model using logistic regression.** A "+" denotes coefficients that are both significant and positive. The symbol ✓ indicates correct identification of the corresponding reaction, respectively.

| Reactions | Case 1 | | | Case 2 | | |
|---|---|---|---|---|---|---|
| | $A_1$ | $A_2$ | | $A_1$ | $A_2$ | |
| $A_1 + A_2 \to 2A_1$ | + | + | ✓ | + | + | ✓ |
| $A_1 + A_2 \to 2A_2$ | + | + | ✓ | + | + | ✓ |
| $A_1 \to \varnothing$ | + | | ✓ | + | | ✓ |
| $\varnothing \to A_1$ | | | ✓ | | | ✓ |
| $A_2 \to \varnothing$ | | + | ✓ | | + | ✓ |
| $\varnothing \to A_2$ (reference) | | | | | | |

and catalyzed production of $P_1$ and $P_2$. The first pair consists of reactions with rates $\kappa_1$ and $\kappa_3$, while the second pair consists of reactions with rates $\kappa_2$ and $\kappa_4$. As a result, when reactions are classified by stoichiometry alone, both the spontaneous production of $P_1$ or $P_2$ and their catalyzed production by another species are treated as the same reaction.

In the HSR model, we assume that all reaction rate constants are of the same order of magnitude, ensuring all types of reactions occur frequently enough. We then consider two scenarios: one where all reaction rate constants are nonzero (Case 1), and another where the reaction rate constant $\kappa_2$ for the natural production of $P_2$ ($\emptyset \to P_2$) is zero (Case 2).

Fig 2 presents ten simulated trajectories for each case. The left panels show the trajectories of three species in Case 1, while the right panels depict those for Case 2. In Case 2, the overall levels of $P_1$ and $P_2$ are slightly reduced compared to Case 1, due to the absence of natural production of $P_2$.

In Case 1, we set the production of $P_2$ as the reference reaction and perform logistic regression as in the TK model. In this case (results shown in Table 2), all reactants involved in the chemical reactions are correctly identified using 10 simulated trajectories, even though the reference reaction includes both the natural and catalyzed production of $P_2$. Further details are given in Supporting information S4E–S4F Tables.

In Case 2, where natural production of $P_2$ is absent ($\kappa_2 = 0$), the reference reaction consists solely of the catalyzed production of $P_2$, which uses $P_1$ as a reactant. Because this reference reaction shares both the same reactant set ($P_1$) and identical reaction rates with two other reactions, the logistic model encounters an identifiability issue: the substrate abundance is identical across three reactions, making it impossible to distinguish the reaction type from $P_1$ alone. This scenario is intentionally constructed to illustrate how the choice of reference reaction influences the ability to identify the underlying chemical network from trajectory data. For this case, we first analyze a dataset of ten simulated trajectories (Case 2a) and then extend the dataset by adding ten additional trajectories (Case 2b).

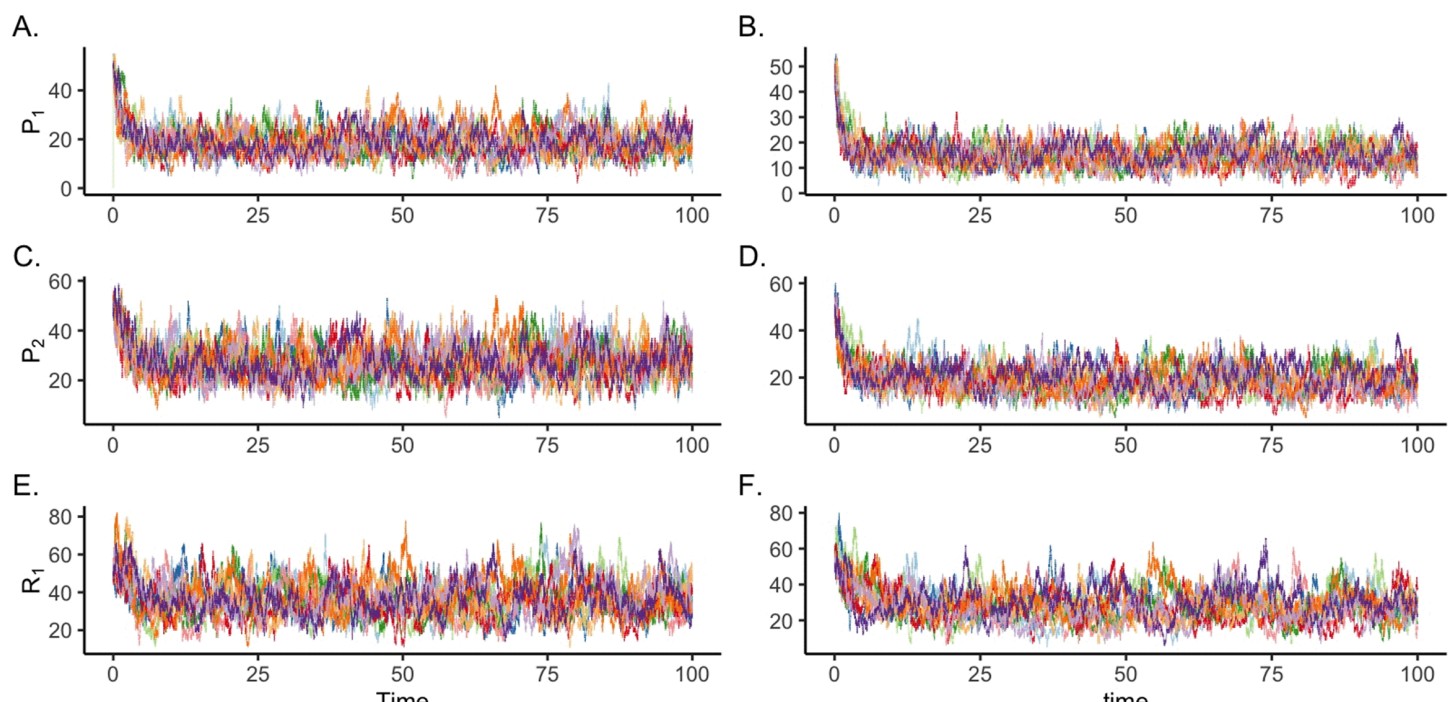

**Fig 2**. **Time-series data of the HSR model.** Ten stochastic simulation trajectories of the heat shock response model are shown, each initialized with $P_1(0) = P_2(0) = R_1(0) = 50$. Panels (A,C,E) depict simulations with nonzero natural production of $P_2$, where $\kappa_1 = \kappa_2 = 10$ and $\kappa_{12} = 0.01$, with all other reaction rate coefficients set to 1. Panels (B,D,F) correspond to simulations with no natural production of $P_2$ ($\kappa_2 = 0$), while all other reaction rate coefficients remain identical to those in Panels (A,C,E).

**Table 2. Species identification in the HSR model using logistic regression.** A "+" indicates that the estimated coefficients are significant and positive. The symbol ✓ denotes correct identification of the corresponding reaction. In Case 2, the absence of natural $P_2$ production ($\kappa_2 = 0$) creates an identifiability issue, as the reference reaction shares its sole reactant $P_1$ and reaction rates with two other reactions. This makes the reaction types indistinguishable from $P_1$ alone and is reflected by the blanks in rows 1 and 7 of the table, which correspond to reactions involving $P_1$.

| Reactions | Case 1 | | | | Case 2a | | | | Case 2b | | | |
|---|---|---|---|---|---|---|---|---|---|---|---|---|
| | $P_1$ | $P_2$ | $R_1$ | | $P_1$ | $P_2$ | $R_1$ | | $P_1$ | $P_2$ | $R_1$ | |
| $P_1 \rightarrow R_1$ | + | | | ✓ | | | | | | | | |
| $P_2 \rightarrow R_1$ | | + | | ✓ | | + | | ✓ | | + | | ✓ |
| $R_1 \rightarrow P_2$ | | | + | ✓ | | | + | ✓ | | | + | ✓ |
| $R_1 \rightarrow 2R_1$ | | | + | ✓ | | | + | ✓ | | | + | ✓ |
| $R_1 + P_2 \rightarrow \varnothing$ | | + | + | ✓ | | + | + | ✓ | | + | + | ✓ |
| $R_1 \rightarrow \varnothing$ | | | + | ✓ | | | + | ✓ | | | + | ✓ |
| $P_1 \rightarrow \varnothing$ | + | | | ✓ | | | | | | | | |
| $P_2 \rightarrow \varnothing$ | | + | | ✓ | | + | | ✓ | | + | | ✓ |
| $\begin{cases} \varnothing \rightarrow P_1, \\ P_2 \rightarrow P_1 + P_2 \end{cases}$ | | + | | ✓ | | + | | ✓ | | + | | ✓ |
| $\begin{cases} \varnothing \rightarrow P_2, \\ P_1 \rightarrow P_1 + P_2 \end{cases}$ (reference) | | | | | $\varnothing \rightarrow P_2$ removed | | | | | | | |

As seen in Table 2 in both scenarios denoted as Case 2a and Case 2b, $P_1$ is not identified as a reactant in the conversion and degradation reactions, as expected. Despite this, the logistic regression correctly identifies the reactants in all remaining reactions in both cases, as summarized in Table 2. Additional details are provided in the Supporting information in S4G–S4J Tables. As shown in S4H Table for Case 2a, several positive $Z$-scores have $P$-values below 0.05 or 0.01; however, the corresponding $Z$-scores are no longer significant in S4J Table for Case 2b. This pattern indicates that larger time-series datasets enhance the reliability of network structure identification.

In addition, most of the estimated regression coefficients for $P_1$ are statistically significant and negative (all except $P_1$:1 and $P_1$:7 in S4F, S4H, and S4J Tables). These negative coefficients indicate that the reference reaction becomes more likely than the reaction under consideration as the abundance of $P_1$ increases. This makes sense because the reference reaction describes the production of $P_2$ catalyzed by $P_1$; higher levels of $P_1$ therefore favor this reaction over its competitors. This interpretation is fully consistent with the logic of logistic regression, which identifies species whose presence increases or decreases the likelihood of a reaction relative to a chosen reference reaction.

**SIR model with demography.** As a final example—used here to illustrate both structural and parameter inference—we consider the classical Susceptible–Infected–Recovered (SIR) reaction network with demographic turnover. This model is widely used for studying epidemic dynamics [23,24]. It includes three types of species: susceptible ($S$), infected ($I$), and recovered ($R$) individuals in a well-mixed population. The model describes the transmission process whereby susceptible individuals become infected through contact with infected individuals, and later transition to the recovered class.

The version of the SIR model considered here incorporates demographic effects by accounting for natural birth and death processes (or, equivalently, immigration and emigration) for all species. The resulting reaction network accounts for the natural turnover of the population and is given by

$$
\begin{aligned}
S + I &\xrightarrow{\beta/n} 2I, \quad \text{(Disease transmission)} \\
I &\xrightarrow{\gamma} R, \quad \text{(Recovery process)} \\
\varnothing &\xrightarrow{\mu n} S, \quad \text{(Natural birth/immigration)} \\
S &\xrightarrow{\nu} \varnothing, \quad I \xrightarrow{\nu} \varnothing, \quad R \xrightarrow{\nu} \varnothing. \quad \text{(Natural death/emigration)}
\end{aligned}
$$

(8)

Here, the parameter $n$ serves as a maximal system size over a pre-specified time window, or more generally, as a scaling parameter that determines the scale at which the population dynamics are considered. The rest of the parameters are as follow: $\beta$ is the rate coefficient for infection and $\gamma$ is for recovery; $\nu$ is the birth rate coefficient for susceptible and $\nu$ is the death rate coefficient for all population. Despite its relative simplicity, this model has been successfully applied and analyzed in the study of several global epidemics—including by some of the present authors—in contexts such as H1N1 [25,26], Ebola [27], COVID-19 [28], and other infectious diseases.

As in the previous sections, we generate synthetic trajectories of the network and apply a logistic regression approach to identify the reactant species involved in each reaction within the extended SIR model. The inflow of susceptibles (representing natural birth or immigration) is used as the reference reaction. Fig 3 illustrates the evolution of the three species counts over time across the 10 selected trajectories. Due to the continuous inflow of susceptible individuals, the overall dynamics differ slightly from those of the basic SIR model (i.e., one defined on a closed population without demography). In particular, both the susceptible and recovered populations exhibit a gradual increase toward the end of the observation period. In contrast, the temporal evolution of the infected population—reflecting epidemic incidence—closely resembles the dynamics observed in the basic SIR model.

The results based on the logistic regression model and data from 10 simulated trajectories are summarized in Table 3. As shown in the table, the reactants involved in all reactions are correctly identified.

## Parameter estimation

After identifying the reactions in a chemical network, estimating the associated parameters is essential for understanding system dynamics and making reliable predictions. While statistical challenges such as stochasticity and partial observation arise in most biochemical systems, epidemic models offer a more tractable setting because population-level data

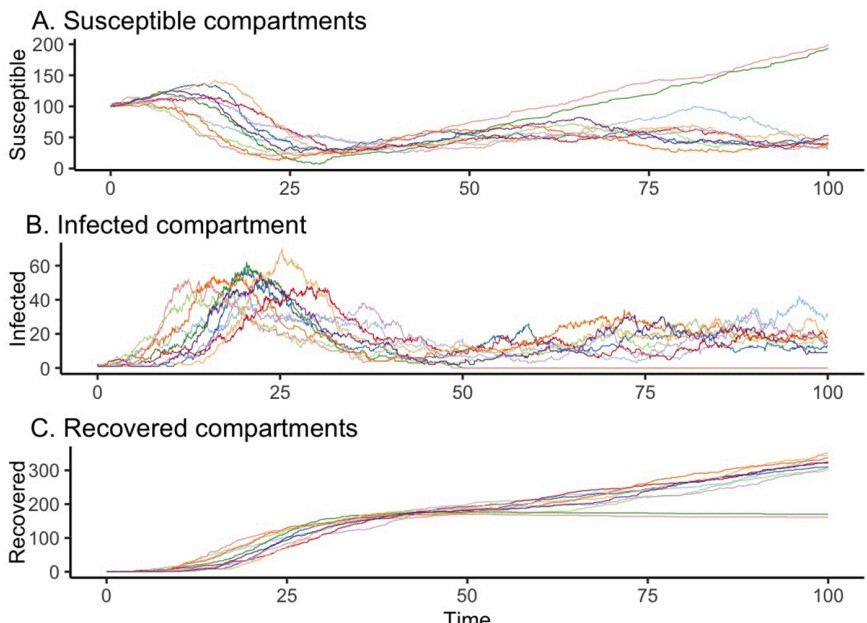

**Fig 3. Time series data from the SIR model with demography.** Ten stochastic simulation trajectories of the SIR model are shown, selected based on sustained infection spread (large outbreak). All runs begin with initial conditions $S(0) = 100$, $I(0) = 1$, and $R(0) = 0$. Reaction rate coefficients are: $\beta/n = 0.004$, $\gamma = 0.2$, $\mu n = 3$, and $\nu = 0.001$.

**Table 3. Species identification in the SIR model with demography using logistic regression.** A "+" indicates that the estimated coefficients are significant and positive. The symbol ✓ denotes correct identification of the corresponding reaction.

| Reactions | Species | | | |
|---|---|---|---|---|
| | *S* | *I* | *R* | |
| $S + I \rightarrow 2I$ | + | + | | ✓ |
| $I \rightarrow R$ | | + | | ✓ |
| $S \rightarrow \varnothing$ | + | | | ✓ |
| $I \rightarrow \varnothing$ | | + | | ✓ |
| $R \rightarrow \varnothing$ | | | + | ✓ |
| $\varnothing \rightarrow S$ (reference) | | | | |

are often readily available. For that reason, we focus on the SIR epidemic model discussed above and show how logistic regression can also be employed for parameter estimation using disease outbreak data.

### Case study: COVID-19 data analysis

Having identified the network structure from synthetic data, we now turn to parameter estimation using observations that approximate real-world conditions. Here, we use the logistic regression framework to estimate the parameters of an SIR model with demography. Our analysis is motivated by epidemic data from the COVID-19 outbreak in Seoul, South Korea, which were previously studied in a different context in [29]. Because these data are protected and cannot be released under current privacy regulations, we instead generated a synthetic dataset inspired by the Seoul outbreak using the procedure outlined in Algorithm S5A in the supporting information.

The dataset includes the time of symptom onset and the time of confirmation for each individual case. Since an infected individual is typically capable of transmitting the virus shortly after symptom onset, we identify the time of infectiousness onset as the time of infection [29]. Upon confirmation of infection, individuals are immediately isolated; thus, we take the date of confirmation to represent the time of removal, as the individual is no longer contributing to transmission.

Using this information, we construct a time-series trajectory of the infected population. Fig 4 illustrates the prevalence from October 17, 2020, to January 24, 2021.

Accordingly, the infection process is modeled by the following stochastic equation [10].

$$I_t = I_0 + Z_1 \left( \int_0^t \frac{\beta}{n} S_u I_u \, \mathrm{d}u \right) - Z_2 \left( \int_0^t (\nu + \gamma) I_u \, \mathrm{d}u \right), \tag{9}$$

where $I_t$ is the count of infected at time $t$, $Z_1$ and $Z_2$ are independent unit Poisson processes and the parameter $n$ represents the *effective population size*. The variable $S_u$ denotes count of susceptible at time $u$, and the rest of the parameters are consistent to the ones in (8). Note that the recovery and degradation of the infected population are represented as a single term in (9), since they follow the same distribution as the infected population in the SIR model. The infected population at each time point is denoted as $\tilde{I} = (i_0, i_1, i_2, \cdots, i_\ell)$, with corresponding observation times $\tilde{T} = (t_0, t_1, t_2, \cdots, t_\ell)$, where $t_j$ is the $j$-th observation time, and we assume all infection events are recorded. We define the infection event indicator as $\tilde{Y} = (y_1, y_2, \cdots, y_\ell)$, where

$$y_j = \begin{cases} 1 & \text{if } i_j - i_{j-1} = 1, \\ 0 & \text{if } i_j - i_{j-1} = -1, \end{cases} \tag{10}$$

for $j = 1, 2, \cdots, \ell$. In this context, $y_j = 1$ corresponds to an infection event, and $y_j = 0$ corresponds to a recovery event. So $\tilde{Y}$ can be the observed response data for the parameter estimation using the logistic regression model.

We can construct the likelihood using the observed response data $\tilde{Y}$ and apply the logistic regression model approach described in the Methods section. At each time $t$, the ratio of infection event probability ($p_t$) to recovery event probability

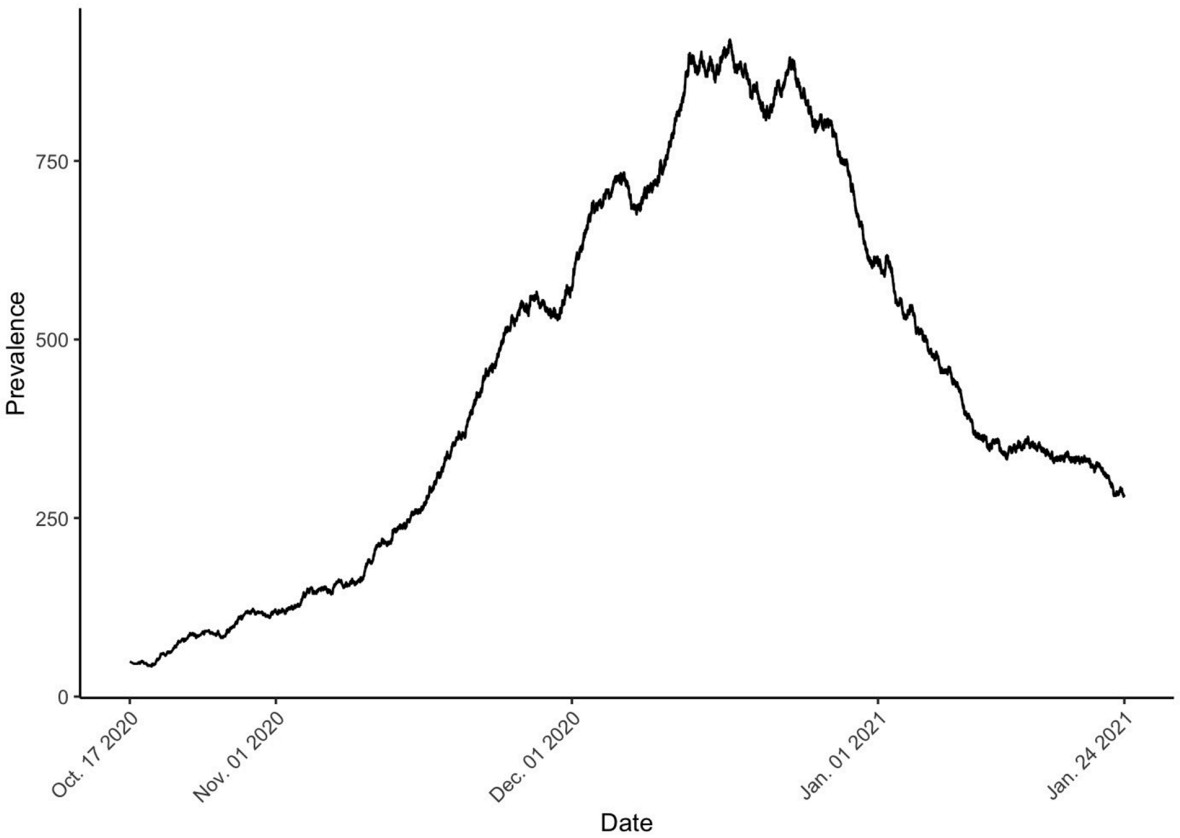

**Fig 4**. Synthetic prevalence of COVID-19 in Seoul, South Korea, from Oct. 17, 2020, to Jan. 24, 2021.

$(1 - p_t)$ is given by

$$\frac{p_t}{1 - p_t} = \frac{\frac{\beta}{n} S_t I_t}{\gamma I_t} = \frac{\beta}{\gamma} \frac{S_t}{n}. \tag{11}$$

Taking the logarithm of (11), we obtain the log-odds as

$$\log\left(\frac{p_t}{1 - p_t}\right) = \log\left(\frac{\beta}{\gamma}\right) + \log\left(\frac{S_t}{n}\right). \tag{12}$$

Since direct observation of $S_t/n$ is not feasible, we approximate it using the law of large numbers relation (see, for instance [15]) $S_t/n \approx s_t$, where $s_t$ is governed by the deterministic SIR model with birth and death processes:

$$
\begin{aligned}
\dot{s}_t &= -\beta s_t \iota_t + \mu - \nu s_t, \\
\dot{\iota}_t &= \beta s_t \iota_t - \gamma \iota_t - \nu \iota_t, \\
\dot{r}_t &= \gamma \iota_t - \nu r_t.
\end{aligned}
\tag{13}
$$

The initial conditions are $s_0 = 1$, $\iota_0 = \rho$, and $r_0 = 0$. Note that the parameters are matching the stochastic model (8). Given that migration rates in Seoul exceed natural birth and death rates, we model the combined effects of immigration

and natural births as a single effective birth rate ($\mu$), and similarly combine emigration and natural deaths into an effective death rate ($\nu$). These demographic rates are calibrated using the net population movement rate during the observation period, which is approximately (based on the census data) $0.013\,\text{day}^{-1}$. Assuming that the total population of Seoul is approximately $10^7$, and only a fraction $n$ of the population is susceptible and may participate in infection events, we estimate the effective birth and death rates as

$$\mu_n = 0.013 \times \frac{n}{10^7}, \quad \nu_n = \frac{0.013}{3} \times \frac{n}{10^7}.$$

Note that $n$ denotes the *effective population size*, which is one of the quantities to be estimated and must therefore be updated during the MCMC procedure. As $n$ evolves throughout the MCMC iterations, the associated birth and death rates, $\mu_n$ and $\nu_n$, also vary accordingly. To simplify the inference process, we fix the effective population size at a chosen time horizon $T > 0$, during which we observe two types of events: infections and recoveries.

Given the total number of observed infections, we approximate the numerical value of the parameter $n$ as the mean of a random variable $N$ drawn from a negative binomial distribution

$$N \sim \text{NegBinomial}(n_I, \tau_T), \tag{14}$$

where $n_I$ denotes the total number of observed infections, and $\tau_T$ represents the probability that a susceptible individual becomes infected by time $T$, accounting for demographic events. The value of $n_I$ is computed as $\sum_{j=1}^{\ell-1} y_j$, representing the cumulative number of infections up to time $T$.

The probability $\tau_T$ is calculated (see (S5-1)) as:

$$\tau_T = \frac{1 - s_T + \mu T - \int_0^T \nu s_u \,\mathrm{d}u}{1 + \mu T - \int_0^T \nu s_u \,\mathrm{d}u}.$$

This definition of $\tau_T$ ensures that (14) yields an estimate of the effective population size that adjusts for birth and death events over the observation window.

According to the log-odds of infection occurrence given in (12), we may formulate a logistic regression model as:

$$\log\left(\frac{P(Y_t = 1)}{1 - P(Y_t = 1)}\right) = \alpha + \text{offset}(\log s_t), \tag{15}$$

where $Y_t$ is a binary indicator of an observed infection event at time $t$, $s_t$ is the susceptible fraction obtained from the solution of the ODE system (13), and $\alpha$ is the intercept of the logistic regression model.

We define the vector of unknown parameters to be estimated as

$$\theta = (\beta, \gamma, \rho), \tag{16}$$

and the corresponding likelihood function is given by (see (4)):

$$L(\theta, n) = \prod_{j=1}^{\ell} P(Y_{t_j} = 1)^{y_j} (1 - P(Y_{t_j} = 1))^{(1-y_j)}.$$

Note that when $Y_{t_j} = y_j$, the concentration of the susceptible population corresponds to $s_t = s_{t_j}$ for $j = 1, 2, \cdots, \ell - 1$. The prior distributions for the parameters vector $\theta$ in (16) are defined independently as

$$
\begin{aligned}
\beta &\sim \text{Gamma}(10^{-3}, 10^{-3}) \equiv f(\beta), \\
\gamma &\sim \text{Gamma}(0.282 \times 10^4, 10^4) \equiv g(\gamma), \\
\rho &\sim \text{Beta}(1, 1) \equiv h(\rho).
\end{aligned}
\tag{17}
$$

Since we have the time of infection and the time of recovery, we can directly estimate the infectious period. So we assigned $\gamma$ an informative prior using the information about the mean infectious period. According to (5), the posterior distribution then satisfies

$$
q(\theta, n) \quad \propto \quad L(\theta, n) \, f(\beta) \, g(\gamma) \, h(\rho).
\tag{18}
$$

Finally, since the posterior distribution in (18) is complex and lacks a closed-form expression, we employ Markov Chain Monte Carlo (MCMC) methods discussed earlier to perform Bayesian estimation of the parameters $\theta$. Specifically, we use the Metropolis–Hastings algorithm within a Gibbs sampler [21], incorporating an offset computed from the solution of the deterministic SIR model with birth and death processes given in (13). The proposed MCMC procedure is outlined in Algorithm 1.

**Algorithm 1 MCMC algorithm for Bayesian logistic regression.**

```
1: Initialize all parameters (θ, n) based on the prior distributions in (17). Initialize the effective
   population size as n = 3 × 10⁴.
2: Solve the system of ODEs in (13) using the current values of (θ, n) to obtain sₜ.
3: Draw samples of θ from their posterior (18) using the Robust Adaptive Metropolis (RAM) algo-
   rithm [19]. Update the intercept α in the logistic model (15).
4: Update the effective population size n sampled from the negative binomial distribution (14) and
   nₗ.
5: Repeat steps 2-4 until convergence.
```

Using this algorithm, we run 4,000 iterations and remove the first half of the iterations as a burn-in set. The last 2000 iterations of the Metropolis–Hastings sampler are used to estimate all parameters, including the effective population size $n$. As shown in S1 Fig, the MCMC simulation quickly reaches stationarity, and all four parameters exhibit good mixing behavior.

Table 4 summarizes the MCMC simulation results for the Bayesian inference. For all parameters, the posterior means and medians are similar, indicating that the posterior distributions are approximately symmetric and that the credible intervals are relatively narrow. Additional diagnostics for the MCMC simulation are provided in the appendix. S1 Fig shows the trace plots for the posterior samples of the parameters, demonstrating rapid convergence to stationarity. S2 Fig displays

**Table 4. Summary statistics for the parameters $\beta$, $\gamma$, $\rho$, and the effective population size (number of individuals at risk of exposure) $n$.**

| Statistic | $\beta$ | $\gamma$ | $\rho$ | Eff. pop. size $n$ |
|---|---|---|---|---|
| Min. | 0.3339 | 0.2671 | 0.000129 | 21,896 |
| 1st Qu. | 0.3558 | 0.2782 | 0.000793 | 26,311 |
| Median | 0.3635 | 0.2826 | 0.001217 | 27,737 |
| Mean | 0.3636 | 0.2824 | 0.001514 | 27,732 |
| 3rd Qu. | 0.3708 | 0.2864 | 0.002131 | 29,147 |
| Max. | 0.3935 | 0.2980 | 0.005205 | 35,741 |

histograms of the posterior samples for each parameter, along with paired scatter plots. With the exception of $\rho$, all histograms are approximately symmetric. Some linear relationships are evident between $\beta$ and $\gamma$, as well as between $\beta$ and the effective population size, but these remain within acceptable limits.

The posterior mean of the basic reproduction number, $R_0 = \beta/\gamma$, is estimated to be 1.288. This estimate closely aligns with previous findings for Seoul [30]. The last column of Table 4 summarizes the estimates of the effective population size, $n$, which is approximately 30,000 during the observation period in Seoul.

In Fig 5, the observed prevalence, the number of infected individuals (black line) is compared to the estimated mean number (blue line) obtained from the Bayesian estimation results. This curve is the mean of the trajectories by solving the SIR model using the set of posterior samples of $\beta$, $\gamma$, $\rho$, and $n$. The red line represents the estimated infection curve, which was generated by solving the SIR model with birth and death processes, using the posterior means of the estimated parameters: $\beta$, $\gamma$, $\rho$, and $n$. Overall, the estimated trend closely follows the observed values. The shaded area represents the 95% credible band for the fitted prevalence curve, which covers most of the observed prevalence. The sharp rise in infections during the early phase of the outbreak is well captured by the model. However, the estimated peak occurs slightly earlier than the actual peak observed in the data.

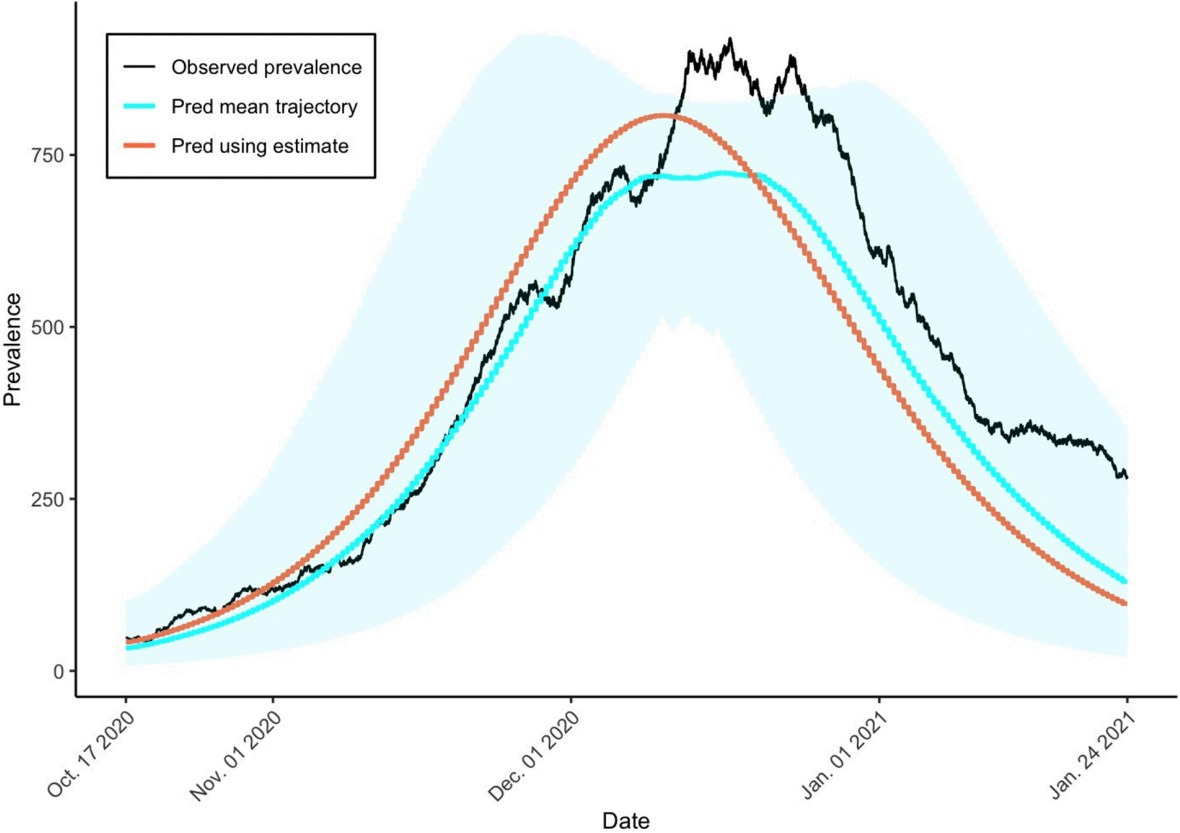

**Fig 5. Comparison of observed and estimated infection prevalence.** The black line shows the observed number of infected individuals over time. The blue line represents the estimated mean infection prevalence, computed as the average of trajectories obtained by solving the SIR model using posterior samples of $\beta$, $\gamma$, $\rho$, and $n$. The red line shows the infection curve generated by solving the SIR model with demography, using the posterior means of the estimated parameters.

## Summary and discussion

Assuming that all chemical reactions follow the law of mass action in a well-mixed environment, we modeled a stochastic chemical reaction network as a continuous-time Markov jump process. Constructing such a model presents two primary challenges: identifying the underlying network structure and inferring its parameters.

To address these challenges, we developed a likelihood-based approach—novel, to the best of our knowledge—that uses logistic regression as its computational engine. In essence, our method provides a scalable variant of the likelihood frameworks considered in [8], but focuses on the embedded discrete Markov chain induced by the stoichiometric vectors. The scalability arises because logistic regression admits highly efficient algorithms for parameter estimation, even in very large systems (i.e., networks with many reactions), provided standard nondegeneracy conditions for the model are satisfied.

This framework is particularly well suited for analyzing open networks that include birth reactions, where species may enter the system from an external source. When full time-series data are available—including molecular counts and reaction events for all species—the logistic-likelihood formulation can be used to identify the network structure, specifically the set of reactants involved in each reaction. By designating a production reaction as the reference category, we may fit a multinomial logistic model to the observed reaction events where the species with statistically significant positive coefficients are interpreted as reactants, since increases in their molecular counts raise the probability of the corresponding reaction occurring.

The reliability of this reactant-identification step depends on the choice of significance thresholds (e.g., $P$–value cutoffs) and on the transformation $z$ applied to the species counts, both of which can substantially affect accuracy. Because the logistic model may involve a large number of potential coefficients, the significance threshold often needs to be more stringent than the conventional value of 0.05. In practice, a suitable cutoff can often be chosen by visually inspecting the distribution of the $P$–values or $Z$-scores; in our analysis, this pragmatic strategy achieved a reasonable balance between strictness and interpretability. With regard to the choice of data transformation $z$, we found across our experiments that the identity transformation was considerably less effective than the logarithmic transformation: the former consistently required more data to recover the correct network structure than the latter. This is perhaps not surprising, as the log transform also aligns with the exact discrete likelihood model.

For parameter estimation, once the reactants for each reaction have been identified, we specify the corresponding propensity functions—typically using a law-of-mass-action form—and estimate the remaining parameters through a second logistic-regression step and Bayesian inference. The Bayesian approach helps absorb some imperfections in the data, such as occasional gaps or unmeasured variables, and generally leads to more stable estimates.

To show how the method can be applied when part of the network is unobserved, we considered synthetic epidemic data modeled after COVID-19 infection time series from the Greater Seoul area. Here we used an extended SIR model with demographic turnover. Because only the number of infected individuals is observed, we introduced an offset term in the logistic regression to represent the unobserved susceptible and recovered populations. The offset was approximated by the deterministic SIR system obtained as the large-population limit of the stochastic model.

Using this modified logistic regression, we built a Bayesian estimation scheme that alternates MCMC sampling with numerical solutions of the SIR ODEs. This allowed us to estimate the infection and recovery rates, the effective population size, and the initial infection level.

A technical difficulty is that the effective population size influences the ODE solution and thus the offset term. We addressed this by solving the ODEs within each MCMC iteration so that the offset is updated together with the parameters. This yielded a stable estimation procedure for partially observed epidemic systems.

The methods proposed here have several important limitations. First, the approach assumes a one-to-one correspondence between reaction types and stoichiometric vectors, as well as the presence of an inflow reaction that can serve as a reference in the logistic model. In addition, structure identification requires full time-series data for all species, a

condition that is rarely met in practice. Parameter estimation is more flexible and can tolerate partial observations, but it still relies on an external deterministic model to construct the offset; without such information, the problem may become unidentifiable [8]. Despite these limitations, our results suggest that logistic-regression-based likelihood methods can extract useful mechanistic information from noisy and incomplete systems. Further refinements may broaden their applicability in biological, ecological, and social settings, where full observability is uncommon. Extending the approach to systems with more substantial missing data remains an important direction for future work.

## Supporting information

**S1 Fig. Trace plots of the posterior sample of the parameters $\theta = (\beta, \gamma, \rho)$ and effective population size $n$.**
(TIFF)

**S2 Fig. Posterior paired scatter plots and histograms of the parameters.** The off-diagonal terms represent the scatter plots between each parameters and the diagonal terms are the histogram of the parameters.
(TIFF)

**S3 Fig. Histogram of $Z$-scores for the TK model under symmetric reaction rates (Case 1).** The red vertical line denotes the threshold $Z$-score 3.09, corresponding to the criterion for coefficient significance ($P < 0.001$).
(TIFF)

**S4 Tables. Species identification and multinomial logistic regression model fitting tables.** Details of the results on logistic regression fitting to the modes discussed in the paper.
(PDF)

**S5 Supplement. Additional Model Details.** Additional calculations and results supporting the paper's conclusions.
(PDF)

## Acknowledgments

The authors gratefully acknowledge the organizers of the *Workshop on Chemical Reaction Network Theory* held at POSTECH (Pohang University of Science and Technology) in the Republic of Korea in July 2024, where this project was initiated. The authors also thank Prof. Enrico Bibbona for his insightful comments on the first version of the manuscript.

## Author contributions

**Conceptualization:** Hye-Won Kang, Grzegorz A Rempala.

**Data curation:** Boseung Choi, Hye-Won Kang.

**Formal analysis:** Boseung Choi.

**Methodology:** Grzegorz A Rempala.

**Project administration:** Hye-Won Kang.

**Software:** Boseung Choi.

**Writing – original draft:** Boseung Choi, Hye-Won Kang, Grzegorz A Rempala.

**Writing – review & editing:** Grzegorz A Rempala.

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
