## [Decision Letter · Decision Letter 0]

22 Oct 2025

PONE-D-25-49199Inferring structure and parameters of stochastic reaction networks with logistic regressionPLOS ONE

Dear Dr. Rempala,

Thank you for submitting your manuscript to PLOS ONE. After careful consideration, we feel that it has merit but does not fully meet PLOS ONE’s publication criteria as it currently stands. Therefore, we invite you to submit a revised version of the manuscript that addresses the points raised during the review process.

We look forward to receiving your revised manuscript.

Kind regards,

Paolo Cazzaniga

Academic Editor

PLOS ONE

Journal Requirements:

3. Please note that your Data Availability Statement is currently missing the repository name and/or the DOI/accession number of each dataset OR a direct link to access each database. If your manuscript is accepted for publication, you will be asked to provide these details on a very short timeline. We therefore suggest that you provide this information now, though we will not hold up the peer review process if you are unable.

4. Please ensure that you refer to Figure 6 in your text as, if accepted, production will need this reference to link the reader to the figure.

5. Please include a caption for figure 6.

6. We notice that your supplementary tables are included in the manuscript file. Please remove them and upload them with the file type 'Supporting Information'. Please ensure that each Supporting Information file has a legend listed in the manuscript after the references list.

Additional Editor Comments:

The manuscript submitted by the authors discusses an interesting and relevant problem for the community of researchers involved in the inference and optimization of stochastic reaction networks.

I agree with the points raised by one of the reviewers, and I therefore recommend a major revision of this work.

Besides the points highlighted by the reviewers, I suggest to better discuss the scalability of the proposed method to understand its actual applicability.

Reviewers' comments:

Reviewer's Responses to Questions

**Comments to the Author**

1. Is the manuscript technically sound, and do the data support the conclusions?

Reviewer #1: Yes

Reviewer #2: Partly

2. Has the statistical analysis been performed appropriately and rigorously?

Reviewer #1: Yes

Reviewer #2: N/A

3. Have the authors made all data underlying the findings in their manuscript fully available?

Reviewer #1: Yes

Reviewer #2: No

4. Is the manuscript presented in an intelligible fashion and written in standard English?

Reviewer #1: Yes

Reviewer #2: Yes

5. Review Comments to the Author

Reviewer #1: In this paper, the authors focus on identifying network structure and parameters in chemical reaction networks. Through stochastic models, it demonstrates that logistic regression can recover full network structures and applies the approach to estimate SIR model parameters for the COVID-19 outbreak in Greater Seoul. In addition, by combining Bayesian regression with differential equation models, the method enables robust inference from partially observed time-series data.

Overall, the paper presents interesting and novel. I recommend that the manuscript be considered for publication after minor revisions.

Reviewer #2: You could cosider my reccomendation as halfway between rejection and Major revisions.

If the author can answer my concerns, the are welcome to (re) submit a new version, but the work required would be important.

6. PLOS authors have the option to publish the peer review history of their article (what does this mean?). If published, this will include your full peer review and any attached files.

Reviewer #1: No

Reviewer #2: **Yes:** Enrico Bibbona

---

## [Author Response · Author response to Decision Letter 1]

8 Dec 2025

the response to reviewers is attached in separate file

---

## [Decision Letter · Decision Letter 1]

11 Jan 2026

Inferring structure and parameters of stochastic reaction networks with logistic regression

PONE-D-25-49199R1

Dear Dr. Rempala,

We’re pleased to inform you that your manuscript has been judged scientifically suitable for publication and will be formally accepted for publication once it meets all outstanding technical requirements.

Kind regards,

Paolo Cazzaniga

Academic Editor

PLOS One

Reviewers' comments:

Reviewer's Responses to Questions

**Comments to the Author**

1. If the authors have adequately addressed your comments raised in a previous round of review and you feel that this manuscript is now acceptable for publication, you may indicate that here to bypass the “Comments to the Author” section, enter your conflict of interest statement in the “Confidential to Editor” section, and submit your "Accept" recommendation.

Reviewer #1: All comments have been addressed

Reviewer #2: All comments have been addressed

2. Is the manuscript technically sound, and do the data support the conclusions?

Reviewer #1: Yes

Reviewer #2: Yes

3. Has the statistical analysis been performed appropriately and rigorously?

Reviewer #1: Yes

Reviewer #2: Yes

4. Have the authors made all data underlying the findings in their manuscript fully available?

Reviewer #1: Yes

Reviewer #2: Yes

5. Is the manuscript presented in an intelligible fashion and written in standard English?

Reviewer #1: Yes

Reviewer #2: Yes

6. Review Comments to the Author

Reviewer #1: (No Response)

Reviewer #2: The present form with corrections and detailed explanations is a huge improvement. The manuscript can now be published as is.

7. PLOS authors have the option to publish the peer review history of their article (what does this mean?). If published, this will include your full peer review and any attached files.

Reviewer #1: No

Reviewer #2: **Yes:** Enrico Bibbona

---

## [Editor Report · Acceptance letter]

PONE-D-25-49199R1

PLOS One

Dear Dr. Rempala,

I'm pleased to inform you that your manuscript has been deemed suitable for publication in PLOS One. Congratulations! Your manuscript is now being handed over to our production team.

Kind regards,

on behalf of

Dr. Paolo Cazzaniga

Academic Editor

PLOS One